# Gnotobiotic rainbow trout (*Oncorhynchus mykiss*) model reveals endogenous bacteria that protect against *Flavobacterium columnare* infection

**David Pérez-Pascual[1]\*, Sol Vendrell-Fernández[1], Bianca Audrain[1], Joaquín Bernal-Bayard[1¤], Rafael Patiño-Navarrete[2], Vincent Petit[3], Dimitri Rigaudeau[4], Jean-Marc Ghigo[1]\***

**1** Unité de Génétique des Biofilms, Institut Pasteur, UMR CNRS2001, Paris, France, **2** Ecologie et Evolution de la Résistance aux Antibiotiques, Institut Pasteur-APHP University Paris Sud, Paris, France, **3** AQUALANDE, Gué de Bern, Pissos, France, **4** Unité Infectiologie Expérimentale Rongeurs et Poissons, INRAE, Université Paris-Saclay, Jouy-en-Josas, France

¤ Current address: Departamento de Genética, Facultad de Biología, Universidad de Sevilla, Sevilla, Spain
\* david.perezpascual@pasteur.fr (DPP); jmghigo@pasteur.fr (JMG)

**Data Availability Statement:** The genome of Flavobacterium sp. 4466 was deposited at

## Abstract

The health and environmental risks associated with antibiotic use in aquaculture have promoted bacterial probiotics as an alternative approach to control fish infections in vulnerable larval and juvenile stages. However, evidence-based identification of probiotics is often hindered by the complexity of bacteria-host interactions and host variability in microbiologically uncontrolled conditions. While these difficulties can be partially resolved using gnotobiotic models harboring no or reduced microbiota, most host-microbe interaction studies are carried out in animal models with little relevance for fish farming. Here we studied host-microbiota-pathogen interactions in a germ-free and gnotobiotic model of rainbow trout (*Oncorhynchus mykiss*), one of the most widely cultured salmonids. We demonstrated that germ-free larvae raised in sterile conditions displayed no significant difference in growth after 35 days compared to conventionally-raised larvae, but were extremely sensitive to infection by *Flavobacterium columnare*, a common freshwater fish pathogen causing major economic losses worldwide. Furthermore, re-conventionalization with 11 culturable species from the conventional trout microbiota conferred resistance to *F. columnare* infection. Using mono-re-conventionalized germ-free trout, we identified that this protection is determined by a commensal *Flavobacterium* strain displaying antibacterial activity against *F. columnare*. Finally, we demonstrated that use of gnotobiotic trout is a suitable approach for the identification of both endogenous and exogenous probiotic bacterial strains protecting teleostean hosts against *F. columnare*. This study therefore establishes an ecologically-relevant gnotobiotic model for the study of host-pathogen interactions and colonization resistance in farmed fish.

European Nucleotide Archive (ENA) databank, accession number: ERS4574862, https://www.ebi.ac.uk/ena/browser/submit.

**Funding:** This work was supported by the Institut Pasteur, the French Government's Investissement d'Avenir program: Laboratoire d'Excellence 'Integrative Biology of Emerging Infectious Diseases' (grant n°. ANR-10-LABX-62-IBEID to J. M.G.), the Fondation pour la Recherche Médicale (grant n°. DEQ20180339185 to J.M.G.). In addition, D.P.-P. was the recipient of an Institut Carnot Pasteur MS post-doctoral fellowship. S.V.-F. was supported by an ERASMUS scholarship and J.B.-B. was the recipient of a long-term post-doctoral fellowship from the Federation of European Biochemical Societies (FEBS). The funders had no role in study design, data collection and analysis, decision to publish, or preparation of the manuscript.

**Competing interests:** I have read the journal's policy and the authors of this manuscript have the following competing interests: a provisional patent application has been filed: "bacterial strains for use as probiotics, compositions thereof, deposited strains and method to identify probiotic bacterial strains" by J.-M.G, D.P.-P. and J.B.-B. The other authors declare no conflict of interest in relation to the submitted work.

## Author summary

The protection provided by host commensal microbiota against pathogens is a long-known phenomenon fostering the notion that introducing beneficial bacteria could reduce or prevent infections. However, the identification of such protective microorganisms is hampered by the poor reproducibility and relevance of current *in vivo* models. We developed procedures to raise germ-free rainbow trout larvae to study the determinants of microbiota-associated resistance to infection. We showed that the fish pathogen *Flavobacterium columnare* rapidly kills infected germ-free but not conventional larvae. We then re-colonized germ-free larvae with bacteria originating from cultured fish microbiota and identified two bacterial species providing full protection against infection. Our approach constitutes a rational and potentially high throughput *in vivo* strategy to study host-pathogen interactions and resistance to infection in fish. The identification of probiotic bacteria protecting rainbow trout and potentially other fish against *F. columnare* could also contribute to improve aquaculture sustainability and health.

## Introduction

As wild fish stock harvests have reached biologically unsustainable limits, aquaculture has grown to provide over half of all fish consumed worldwide [1]. However, intensive aquaculture facilities are prone to disease outbreaks and the high mortality rate in immunologically immature juveniles, in which vaccination is unpractical, constitutes a primary bottleneck for fish production [2–4]. These recurrent complications prompt the prophylactic or therapeutic use of antibiotics and chemical disinfectants to prevent fish diseases [5,6] but may lead to final consumer safety risks, environmental pollution and spread of antibiotic resistance [7]. In this context, the use of bacterial probiotics to improve fish health and protect disease-susceptible juveniles is an economic and ecological sensible alternative to antibiotic treatments [8,9].

Probiotics are live microorganisms conferring health benefits on the host via promotion of growth, immuno-stimulation or direct inhibition of pathogenic microorganisms [10,11]. The native host microbiota plays a protective role against pathogenic microorganisms by a process known as colonization resistance [12,13]. In fish, the endogenous microbial community, whether residing in gastrointestinal tract or in the fish mucus, was early considered as a source of protective bacteria [14–18]. However, selection of probiotic bacteria is often empirical or hampered by the poor reproducibility of *in vivo* challenges, frequently performed in relatively uncontrolled conditions with high inter-individual microbial compositions [15,19].

To improve evidence-based identification of fish probiotics and their efficacy in disease prevention, the use of germ-free (GF) or fully controlled gnotobiotic hosts is a promising strategy [20,21]. In addition to laboratory fish models such as zebrafish (*Danio rerio*) [22–24], several fish species have been successfully reared in bacteria-free conditions to test probiotic-based protection against pathogenic bacteria, including Atlantic cod (*Gadus morhua*) [25], Atlantic halibut (*Hippoglossus hippoglossus*) [26], and European sea bass (*Dicentrarchus labrax*) [19] (for a review, see [27]).

Salmonids, especially rainbow trout (*Oncorhynchus mykiss*) and Atlantic salmon (*Salmo salar*), are economically important species, whose production in intensive farming is associated with increased susceptibility to diseases caused by viruses, bacteria and parasites [28]. Here we studied the probiotic potential of endogenous members of the rainbow trout microbiota to protect against infection by *Flavobacterium columnare*, a fresh-water fish pathogen causing major losses in aquaculture of fish such as Channel catfish, Nile tilapia and salmonids

[29]. We developed a new protocol to rear GF trout larvae and showed that GF larvae were extremely sensitive to infection by *F. columnare*. We then identified two bacterial species originating either from the trout microbiota (a commensal *Flavobacterium* sp.) or the zebrafish microbiota (*Chryseobacterium massiliae*) that fully restored protection against *F. columnare* infection. Our *in vivo* approach opens perspectives for the rational and high throughput identification of probiotic bacteria protecting rainbow trout and other fish against columnaris disease. It also provides a new model for the study of host-pathogen interactions and colonization resistance in a relevant teleostean fish model.

## Results

### Germ-free trout show normal development and growth compared to conventional larvae

To produce microbiologically controlled rainbow trout and investigate the potential protection conferred by endogenous or exogenous bacteria against incoming pathogens, we produced (GF) trout larvae by sterilizing the chorion of fertilized eggs with a cocktail of antibiotics and antifungals, 0.005% bleach and a iodophor disinfection solution. GF eggs were then kept at 16˚C under bacterial-free conditions and both conventional (Conv) and treated eggs hatched spontaneously 5 to 7 days after reception, indicating that the sterilization protocol did not affect the timing of hatching. However, hatching percentage was 72 ± 5.54% for sterilized eggs versus 48.6 ± 6.2% for non-treated. Once hatched, all larvae were transferred into vented-cap cell culture flasks containing fresh sterile water without antibiotics renewed every 48 hours (h). GF and Conv fish relied on their vitellus reserves until 20 days post-hatching (dph) after which they were fed with sterilized fish food powder every 48 h (Fig 1). Sterility tests were performed at 24 h, 7 days and 21 days post-sterilization treatment and before each water change until the end of the experiment (35 dph) (S1 Fig). To test the physiological consequences of raising GF larvae, we compared the growth of Conv and GF larvae reared from the same batch of fertilized eggs and observed no significant difference in standard body length (2.51 ± 0.24 cm *vs*. 2.58 ± 0.21 cm) or weight (1.17 ± 0.20 g *vs*. 1.17 ± 0.10 g) at 35 dph for Conv and GF, respectively (S2 Fig). To compare Conv and GF trout anatomy, we developed an approach combining iDISCO solvent-based method to generate transparent fish tissue and

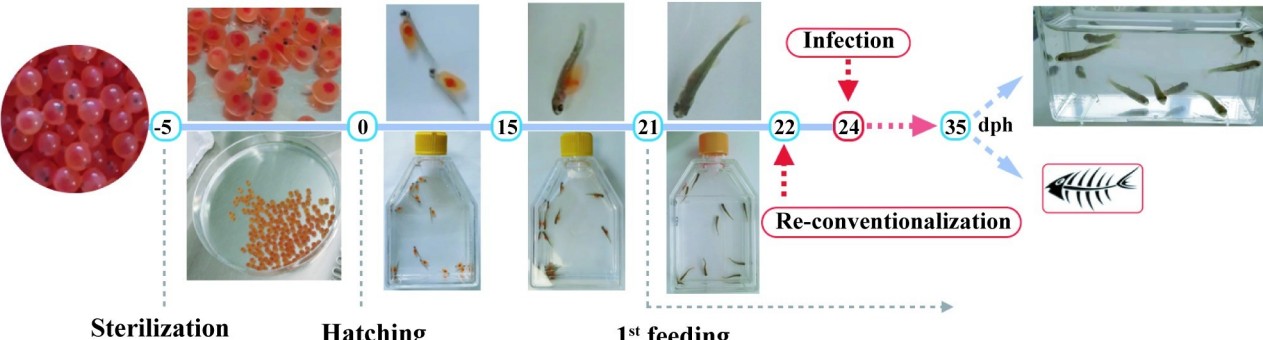

**Fig 1. Protocol used in this study to raise and infect or re-conventionalize germ-free (GF) trout larvae.** Eyed eggs were sterilized 5 days before hatching and kept in sterile, autoclaved mineral water at 16˚C in Petri dishes until hatching. Once hatched (0 day post hatching = dph), rainbow trout larvae were transferred into vented cap cell culture flasks for the duration of the experiment. Larvae were fed every 2 days with sterile powder food from 21 dph until the end of the experiment; water was renewed 30 minutes after feeding. To test the protective effect of potential probiotic strains, larvae were re-conventionalized by one or several commensal bacteria diluted in water at 22 dph. Pathogenic bacteria were added to the water at 24 dph for 24 h and then larvae were washed with fresh sterile water. Survival after infection was monitored twice per day.

lightsheet 3D imaging of the whole body. This analysis did not reveal any anatomical differences at 21 dph, even regarding organs in direct contact with fish microbiota such as gills (Fig 2D and 2I) and intestine (Figs 2C and 2H and S3). No difference was seen in other organs potentially influenced by gut-microbiota such as the brain (Fig 2A and 2F), spleen (Fig 2B and 2G) and head kidney (Fig 2E and 2J) [30]. These results suggested that the natural microbiota had no major macroscopic impact on fish growth, development or anatomy at this stage of rainbow trout development in our rearing conditions.

## Identification of susceptibility to fish pathogens in germ-free but not conventional trout larvae

To identify bacterial pathogens able to infect GF rainbow trout larvae by the natural infection route, we exposed the 24 dph larvae for 24 h to $10^7$ colony forming units (CFU)/ml of several trout bacterial pathogens, including *Flavobacterium psychrophilum* strain THCO2-90, *F. columnare* strain Fc7, *Lactococcus garvieae* JIP 28/99, *Vibrio anguillarum* strain 1669 and *Yersinia ruckeri* strain JIP 27/88 [31]. Larvae were then washed with sterile water, renewing 90% of the infection water three times and kept at 16°C under sterile conditions. Among all tested pathogens, only *F. columnare* strain Fc7 led to high and reproducible mortality of GF trout larvae within 48 h post-exposure (Fig 3). In contrast, Conv larvae reared from non-sterilized eggs survived *F. columnare* strain Fc7 infection under tested conditions (Figs 4A and S4). Histological analysis performed at 25 dph (24 h post infection) on GF and Conv larvae did not show any sign of intestinal damage (S5 Fig). However, we observed an increase in goblet cells number in infected *vs* non-infected GF larvae, whereas Conv infected larvae showed the opposite phenotype when compared to non-infected Conv larvae (S5 Fig).

## Conventional rainbow trout microbiota protects against *F. columnare* infection

Considering the high sensitivity of GF but not Conv trout larvae to infection by *F. columnare* Fc7, we hypothesized that resistance to infection could be provided by some components of the Conv larvae microbiota. To test this, we exposed GF rainbow trout larvae to water from Conv larvae flasks at 22 dph. Re-conventionalized (Re-Conv) rainbow trout larvae survived as well as Conv larvae to *F. columnare* Fc7 infection, whereas those maintained in sterile

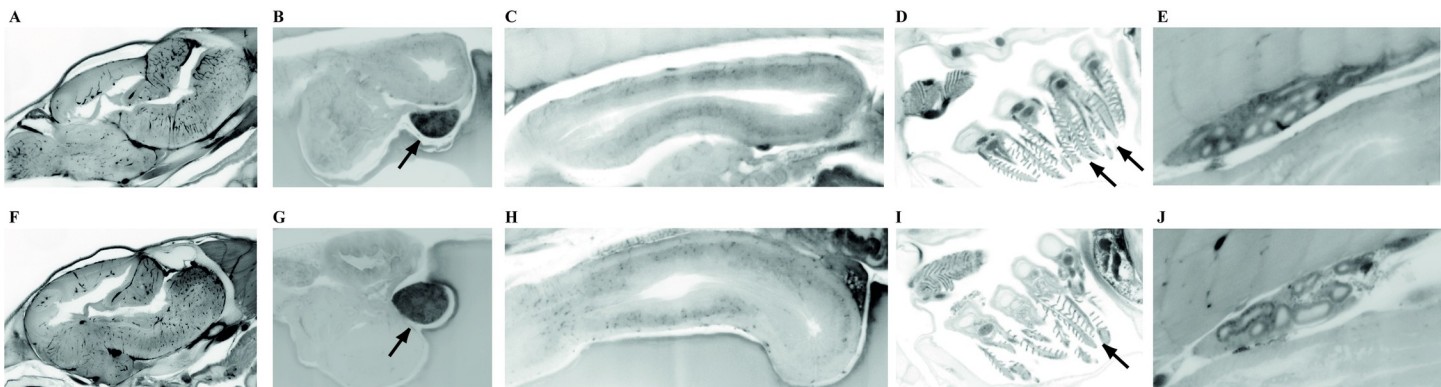

**Fig 2. Anatomical comparison of Conventional (Conv) and GF rainbow trout larvae.** 3D deep imaging of whole trout body corresponding to autofluorescence signal acquired by lightsheet microscopy after novel fish clearing processing. Selected optical sections of 21 dph were presented for Conv (A, B, C D and E) and GF (F, G, H, I and J) rainbow trout larvae. Brain (A and F), spleen (black arrow in B and G), gut (C and H) (see also S3 Fig), gills (black arrows in D and I), and head kidney (E and J). Images representative of two different fish per condition.

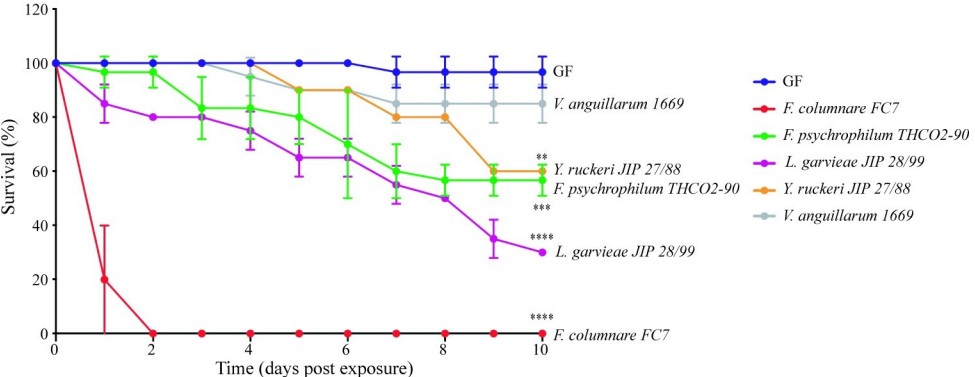

**Fig 3. Survival of GF and Conv rainbow trout larvae infected with different fish pathogens.** Kaplan-Meier graph of GF larvae survival after bath exposure to *F. psychrophilum* strain THCO2-90, *F. columnare* strain Fc7, *L. garvieae* strain JIP 28/99, *V. anguillarum* strain 1669 and *Y. ruckeri* strain JIP 27/88. Mean and SD plot representing average survival percentage of fish for 10 days after exposure to different pathogenic microorganisms. For each condition n = 10 larvae. All surviving fish were euthanized at day 10 post-infection. Asterisks indicate significant difference from non-infected population (**p<0.01; ***p<0.001; ****p<0.0001).

conditions died within the first 48h after infection (Fig 4B). These results suggested that microbiota associated with Conv rainbow trout provide protection against *F. columnare* Fc7 infection. To identify culturable species potentially involved in this protection, we plated bacteria

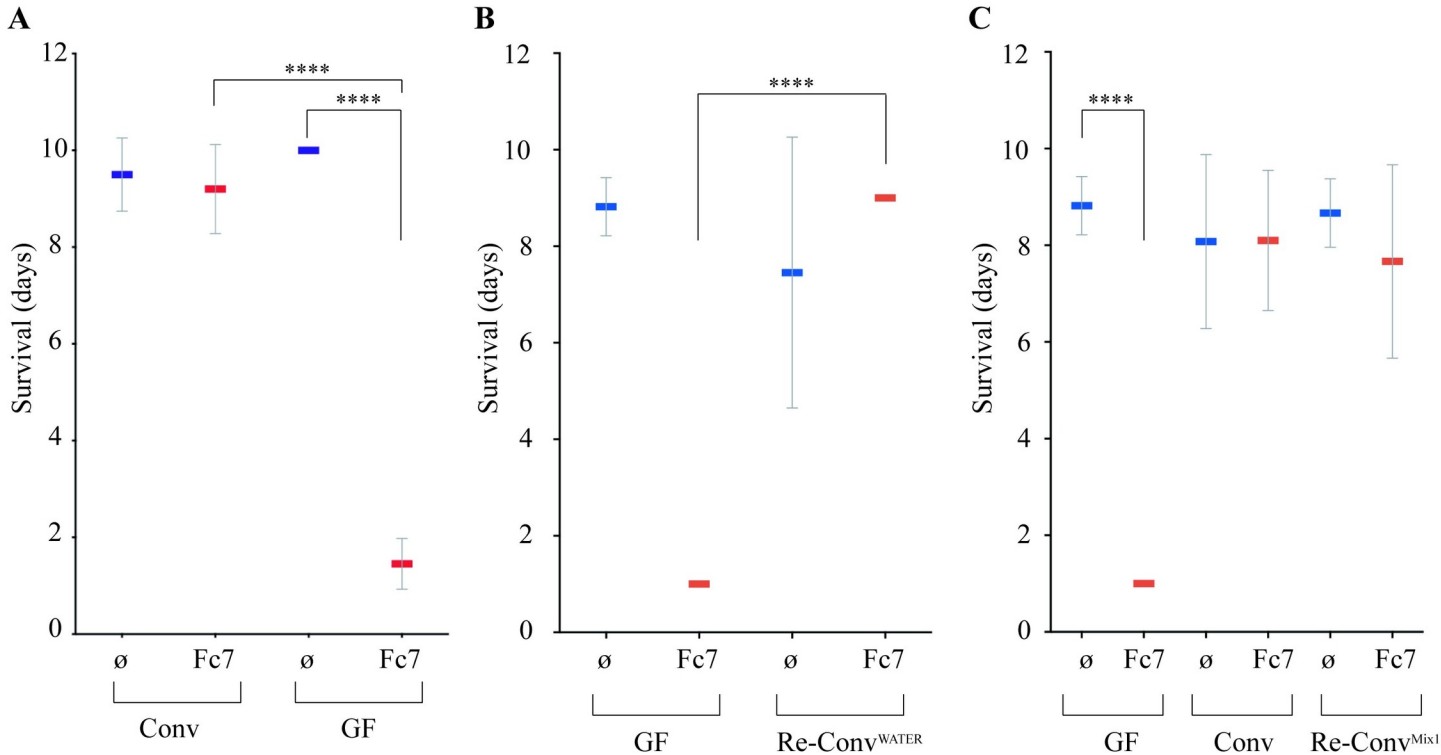

**Fig 4. Survival of re-conventionalized trout larvae against *F. columnare* Fc7 infection. A:** *F. columnare* strain Fc7 kills GF but not Conv rainbow trout. Mean and SD plot representing average day post-infection at which infected fish die. For each condition n = 10 larvae. All surviving fish were euthanized at day 10 post-infection. Asterisks indicate significant difference from non-infected population (****p<0.0001). **B:** GF trout larvae exposed to water used to raise Conv fish at 22 dph show similar survival rates to *F. columnare* infection than Conv trout larvae. **C:** The 11 strains identified from Conv fish microbiota were added to rainbow trout larvae at 22 dph, followed by *F. columnare* infection at 24 dph. This bacterial mixture is able to protect re-conventionalized larvae from infection. For each condition n = 10 larvae. All surviving fish were euthanized at day 10 after infection (****p<0.0001).

**Table 1. The 11 strains isolated from Conv rainbow trout larvae.**

| Bacterial strains isolated from trout microbiota |
| --- |
| *Aeromonas rivipollensis 1* strain 4512 |
| *Pseudomonas helmanticensis* strain 4513 |
| *Aeromonas rivipollensis 2* strain 4514 |
| *Pseudomonas baetica* strain 4515 |
| *Aeromonas hydrophila* strain 4516 |
| *Flavobacterium plurextorum 1* strain 4517 |
| *Acinetobacter* sp. strain 4518 |
| *Flavobacterium plurextorum 2* strain 4519 |
| *Delftia acidovorans* strain 4465 |
| *Flavobacterium* sp. strain 4466 |
| *Pseudomonas* sp. strain 4520 |

recovered from 3 whole Conv rainbow trout larvae at 35 dph on various agar media. 16S rRNA-based analysis of each isolated morphotype led to the identification of 11 different bacterial strains corresponding to 9 different species that were isolated and stored individually (Table 1).

We then re-conventionalized GF rainbow trout larvae at 22 dph with an equiratio mix of all 11 identified bacterial strains (hereafter called Mix11), each at a concentration of $5.10^5$ CFU/ ml. After exposure to *F. columnare* strain Fc7, these Re-Conv$^{Mix11}$ larvae survived as well as Conv fish (Fig 4C), demonstrating that the Mix11 isolated from the rainbow trout microbiota recapitulates full protection against *F. columnare* infection observed in Conv larvae.

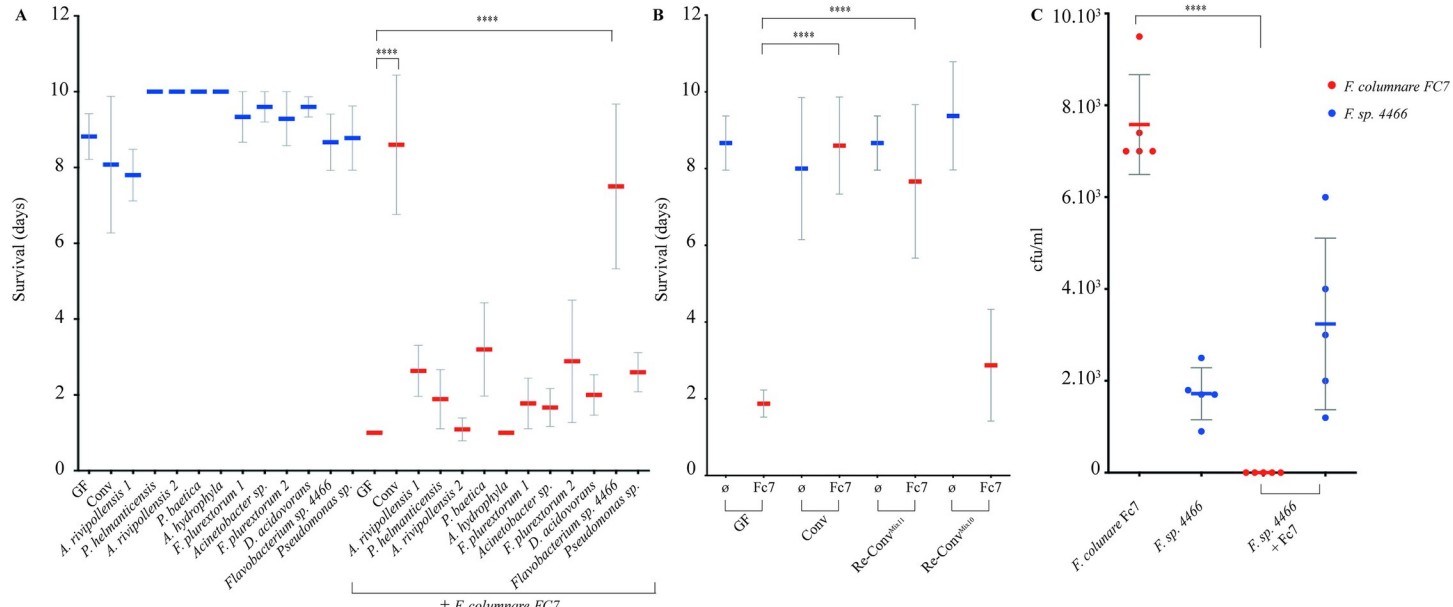

**Fig 5. Protection of GF trout larvae against *F. columnare* infection by individual species isolated from the Conv rainbow trout microbiota. A:** The 11 species isolated from Conv fish microbiota (Table 1) were added individually to rainbow trout larvae at 22 dph, followed by *F. columnare* Fc7 infection at 24 dph. From the 11 different strains, only *Flavobacterium* sp. strain 4466 protected re-conventionalized larvae from infection. **B:** Mix11, Mix10 (mix of all identified strain with the exception of *Flavobacterium* sp. strain 4466), were added to rainbow trout larvae at 22 dph, followed by *F. columnare* infection at 24 dph. Mix11 protected re-conventionalized larvae from infection, whereas Mix10 did not. For each condition n = 10 larvae. All surviving fish were euthanized at day 10 after infection. C: CFU/mL recovered from dissected intestines from GF fish exposed to *F. columnare* Fc7, *Flavobacterium* sp. strain 4466 or both, 24 hours post-infection. (****p<0.0001).

### Resistance to *F. columnare* infection is conferred by one member of the trout microbiota

To determine whether some individual members of the protective Mix11 could play key roles in infection resistance, we mono-re-conventionalized 22 dph GF trout by each of the 11 isolated bacterial strains at $5.10^5$ CFU/ml followed by challenge with *F. columnare* Fc7. We found that only *Flavobacterium* sp. strain 4466 restored Conv-level protection, whereas the other 10 strains displayed no protection, whether added individually (Fig 5A) or as a mix (Mix10 in Fig 5B). To evaluate the colonization of gastrointestinal tract by *Flavobacterium* sp. strain 4466 and/or *F. columnare* Fc7, we plated intestines from mono-reconventionalized fish 24 hours after infection in TYES agar after dissection in sterile conditions. Interestingly, whereas both *Flavobacterium* sp. strain 4466 and *F. columnare* Fc7 were able to successfully colonize the gut of mono-exposed rainbow trout (Fig 5C), we only detected *Flavobacterium* sp. after trout's infection by *F. columnare* Fc7 (Fig 5C), suggesting a potential competition between both bacterial species. Consistently, although cell-free spent supernatant of *Flavobacterium* sp. strain 4466 showed no inhibitory activity against *F. columnare* Fc7 in an overlay assay (Fig 6A), *Flavobacterium* sp. strain 4466 colony growth inhibited the growth of *F. columnare* Fc7 (Fig 6B) and of all tested *F. columnare* strains (Fig 6C), suggesting a potential contact dependent inhibition. We identified a cluster of 12 genes potentially associated to this phenotype in the *Flavobacterium* sp. strain 4466 genome (*tssB, tssC, tssD, tssE, tssF, tssG, tssH, tssI, tssK, tssN, tssP* and *tssQ*) characteristic of type 6 secretion system (T6SS), T6SS[iii], a contact-dependent antagonistic system only present in phylum *Bacteroidetes* [32]. To improve the taxonomic identification of the protective *Flavobacterium* isolated from the trout larvae microbiota, we performed whole genome sequencing followed by Average Nucleotide Identity (ANI) analysis. We determined that despite similarity with *Flavobacterium spartansii* (94.65%) and *Flavobacterium tructae* (94.62%), these values are lower than the 95% ANI needed to identify two organisms as the same species [33]. Furthermore, full-length 16S rRNA and *recA* genes comparisons also showed high similarity with *F. spartansii* and *F. tructae*, however, the obtained values were also below the 99% similarity threshold required to consider that two organisms belong to the same species (S1 Table). Similarly, a maximum likelihood based phylogenetic tree (S6 Fig) generated from sequences of 15 bacterial strains from the *Flavobacterium* genus revealed that the

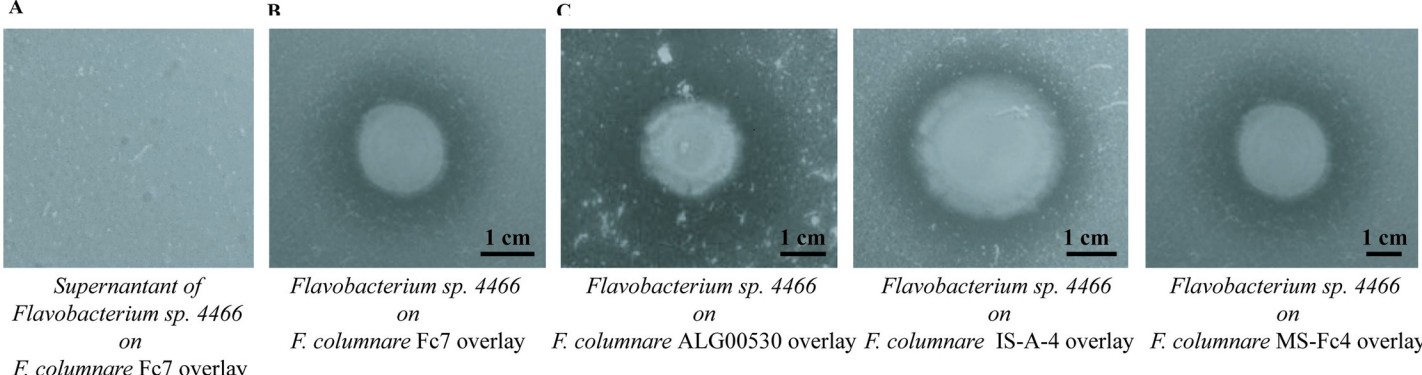

*Supernantant of Flavobacterium sp. 4466 on F. columnare Fc7 overlay*   *Flavobacterium sp. 4466 on F. columnare Fc7 overlay*   *Flavobacterium sp. 4466 on F. columnare ALG00530 overlay*   *Flavobacterium sp. 4466 on F. columnare IS-A-4 overlay*   *Flavobacterium sp. 4466 on F. columnare MS-Fc4 overlay*

**Fig 6. Representative images of *in vitro* growth-inhibition activity of *Flavobacterium* sp. strain 4466 against different virulent *F. columnare* strains. A:** lack of *F. columnare* Fc7 growth-inhibition after adding 5 μl of *Flavobacterium sp*. culture supernatant. **B:** Halo of *F. columnare* FC7 growth inhibition surrounding *Flavobacterium* sp. colony on a *F. columnare* strain Fc7 overlay. **C:** Halo of growth inhibition of *F. columnare* ALG-00-530, IA-S-4, and Ms-Fc-4. The agar overlay technique was performed by spreading *F. columnare* bacterial suspension on soft-agar solution over TYES agar, and then spotting 5 μl of an overnight culture of *Flavobacterium* sp. strain 4466. Incubation was performed at 28°C for 24 h. This experiment was performed in triplicate.

sequence of *Flavobacterium* sp. strain 4466 clustered with sequences of *F. spartansii* and *F. tructae*, but did not allow the identification of *Flavobacterium* sp. strain 4466 at species level.

### Endogenous *Flavobacterium* sp. strain 4466 protects germ-free rainbow trout against infection by different strains of *F. columnare*

To test whether the protective *Flavobacterium* sp. isolated from the Conv rainbow trout microbiome could protect rainbow trout we re-conventionalized GF fish larvae with *Flavobacterium* sp. 48 hours before exposure to four virulent *F. columnare* strains (Fc7, ALG-00-530, IA-S-4, and Ms-Fc-4) belonging to genomovars I and II, and isolated from different geographical origins and host fish species. *Flavobacterium* sp. strain 4466 conferred protection to rainbow trout larvae against all *F. columnare* strains (Fig 7). Therefore, the *Flavobacterium* sp. strain identified from trout Mix11 is a putative probiotic potentially protecting trout from columnaris disease.

### Use of germ-free trout model to validate exogenous probiotics protecting against *F. columnare* infection

To determine whether our GF trout model could be used as a controlled gnotobiotic approach to screen for trout probiotics, we pre-exposed 22 dph GF rainbow trout larvae to *Chryseobacterium massiliae*, a bacterium that does not belong to trout microbiota but was previously shown to protect larval stage and adult zebrafish from infection by *F. columnare* [34]. After 48 h of bath in a *C. massiliae* suspension at $10^5$ CFU/ml, we infected trout larvae with *F. columnare* strains Fc7, ALG-00-530, IA-S-4 and Ms-Fc-4 and observed that *C. massiliae* protected against all tested *F. columnare* pathogens (Fig 8). These results showed that the GF rainbow trout model enables the evaluation of bacterial species, endogenous to trout or not, with probiotic potential against highly virulent *F. columnare* strains.

## Discussion

Although the use of probiotics is a promising approach to improve fish growth and reduce disease outbreaks while limiting chemical and antibiotic treatments [17,35,36], rational and evidence-based procedures for the identification of protective bacteria are limited. Here, we established a controlled and robust model to study trout resistance to infection by bacterial pathogens and to identify trout probiotics in microbiologically controlled conditions using GF and gnotobiotic rainbow trout.

Our gnotobiotic protocol is based on the survival of rainbow trout eggs to chemical sterilization eliminating the microbial community associated to the egg surface. The lower hatching efficiency observed for Conv eggs compared to germ-free eggs could be due to different factors such as physico-chemical parameters modification induced by endogenous microbiota (i.e. dissolved $O_2$ concentration or water pH), or higher susceptibility to opportunistic infections. Similarly to gnotobiotic protocols used for zebrafish [24,37], cod [25] and stickleback (*Gasterosteus aculeatus*) [38], our approach produced larvae that were GF from hatching to 35 dph at 16°C without continued exposure to antibiotics, therefore avoiding possible effects of prolonged antibiotic exposure on fish development [39]. Similarly to GF stickleback larvae at 14 dph [38], we observed no development or growth differences between GF and Conv trout larvae at 21 dph. In contrast, GF sea bass (*D. labrax* L.) larvae grew faster and had a more developed gut compared to conventionally raised larvae [40]. These discrepancies could come from the fact that, in our study and in the GF stickleback study, anatomical analyses were performed before first-feeding, whereas the GF sea bass were already fed when examined [40]. Indeed,

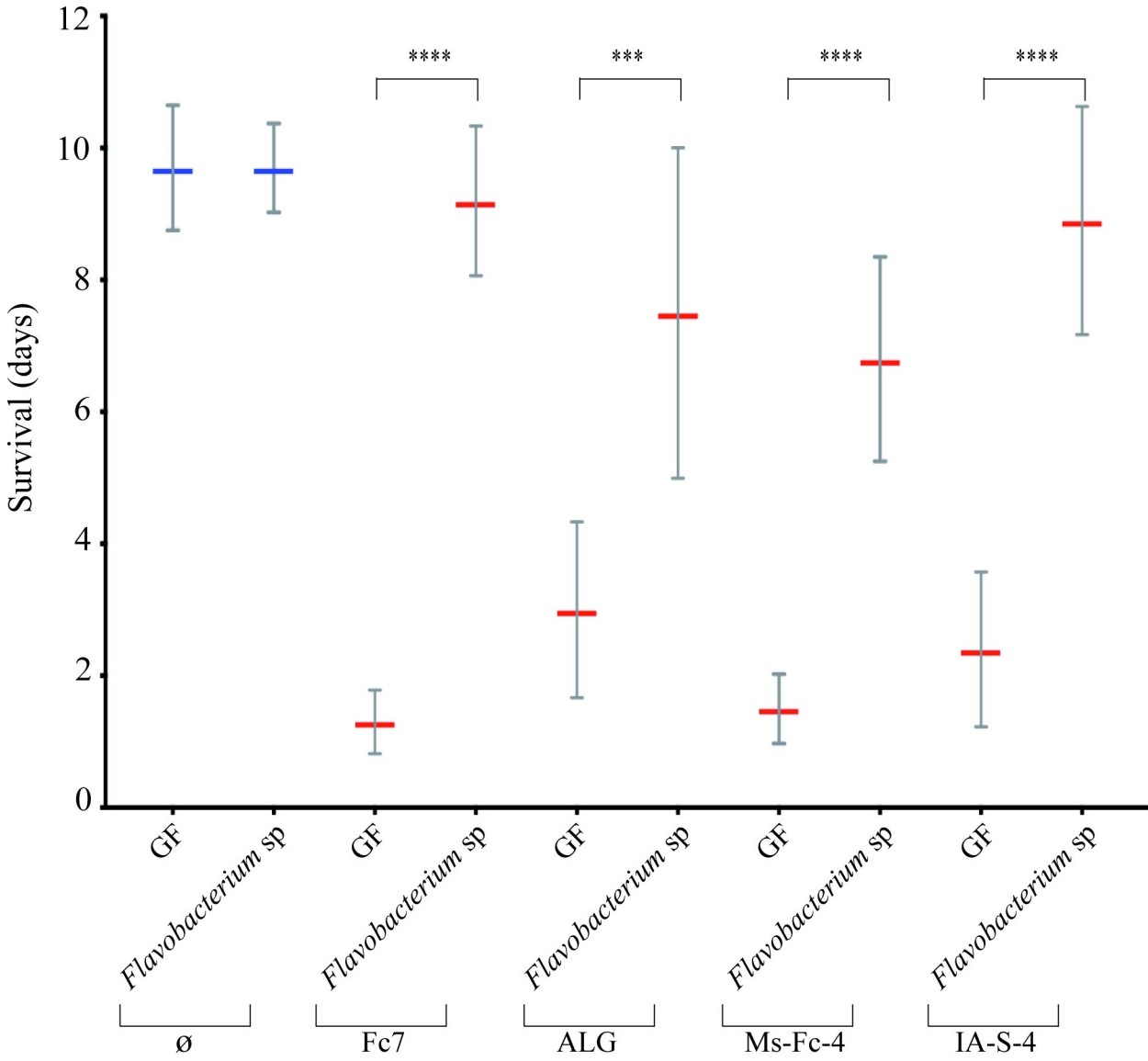

**Fig 7. Strain 4466 provides full protection to gnotobiotic rainbow trout larvae against infection by four strains of *F. columnare*. *Flavobacterium* sp.** Survival of GF trout larvae exposed to *Flavobacterium* sp. strain 4466 48 h before infection with *F. columnare* strains Fc7, IA-S-4, Ms-Fc-4 and ALG-00-530. All *F. columnare* strains rapidly killed GF fish, whereas trout larvae that were re-conventionalized with *Flavobacterium* sp strain 4466 survived to all strains of F. columnare. Mean and SD plot representing average day post-infection at which infected fish died. For each condition n = 10 larvae. All surviving fish were euthanized at day 10. Asterisks indicate significant difference from non-infected population (****p<0.0001).

trout larvae initially acquire nutrients by absorbing their endogenous yolk until the intestinal track is open from the mouth to the vent. We therefore cannot rule out that at later stages of development, when fish begin to rely on external feeding, differences between GF and Conv fish may occur, especially in the structure and size of organs or in body weight. However, the hurdles associated with long-term fish husbandry while keeping effective sterility control, *de facto* limits our approach to relatively short-term experiments on larvae with limited feeding time and low complexity microbiota.

While GF conditions cannot be compared to those prevailing in the wild or used in fish farming [25], our results showed that GF rainbow trout larvae are highly susceptible to *F*.

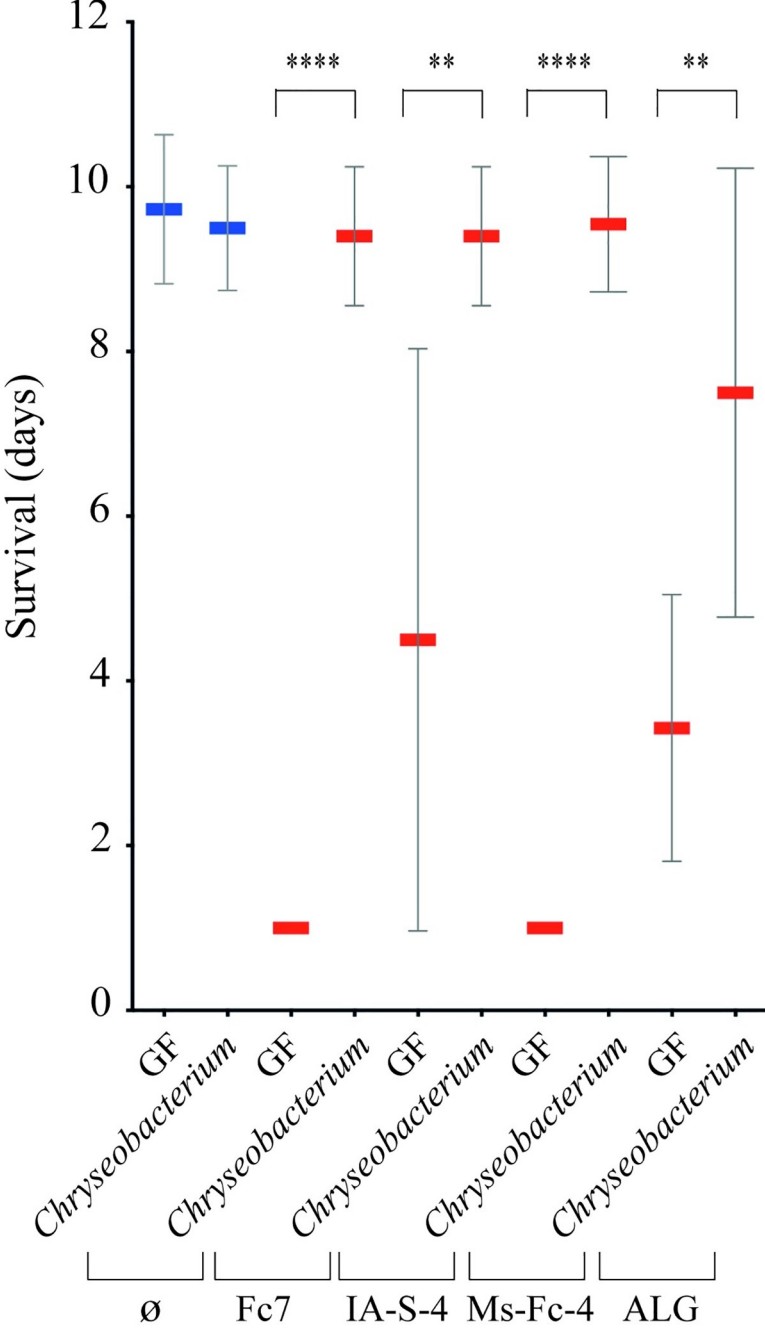

**Fig 8. *C. massiliae* provides trout protection against *F. columnare* infection.** GF trout larvae survival exposed to *C. massiliae* 48 h before infection with *F. columnare* strains Fc7, IA-S-4, Ms-Fc-4 and ALG-00-530. Mean and SD plot representing average day post-infection at which infected fish die. For each condition n = 10 larvae. All surviving fish were euthanized at day 10. Asterisks indicate significant difference from non-infected population (\*\*\*\*p<0.0001; \*\*p<0.01).

*columnare*, the causative agent of columnaris disease affecting many aquaculture fish species [29,41].

Although our histology analysis comparing GF and Conv larvae infected or not by *F. columnare* Fc7 did not show any major sign of inflammation of damage, we observed that the

number of Goblet cells per crypt increases in infected GF larvae and decresed in Conv larvae. A healthy intestine is determined by biological markers such as Goblet cells count, which secrete mucus with bactericidal properties [42]. Interestingly, a significative decrease in the number of Goblet cells was also observed of non-infected GF larvae compared to those of Conv larvae, as previously reported in zebrafish [37]. The absence of stimulating microorganisms in GF larvae could lead to a dysregulated acute immune response after *F. columnare* infection. These results suggest that the microbiota influence cell differentiation (or maturation) in trout gut epithelium [43], potentially affecting for some aspect of the protection against *F. columnare* infection.

Interestingly, our GF rainbow trout larvae model also revealed the protective activity of *C. massiliae*, a potential probiotic bacterium isolated from Conv zebrafish [34], against various *F. columnare* strains from different fish host and geographical origins. These results demonstrate that GF rainbow trout is a robust animal model for the study of *F. columnare* pathogenicity and support *C. massiliae* as a potential probiotic to prevent columnaris diseases in teleost fish other than its original host.

Furthermore, we demonstrated that the relatively simple culturable bacteria isolated from microbiota harbored by Conv trout larvae effectively protect against *F. columnare*. Interestingly, different studies have demonstrated that highly diverse gut communities are more likely to protect the host from pathogens [44,45]. This constitutes the base for the paradoxical negative health effect associated with the massive utilization of antibiotics in aquaculture: the reduction in microbial diversity facilitates colonization by opportunistic pathogens [46]. While this advocates for practices leading to enrichment of fish microbial communities to minimize pathogenic invasions in aquaculture [16], our results demonstrate that resistance to a bacterial pathogen can also be achieved by a single bacterial strain in a low complexity microbiota. Moreover, previous studies of resistance to infection provided by controlled bacterial consortia in gnotobiotic hosts often relied on community composition, rather than individual members of the microbiota [47–50]. We showed that the observed protection in larvae is mainly due to the presence of *Flavobacterium* sp. strain 4466. We cannot exclude, however, that at later developmental stages, the presence of other bacterial species may be needed for more efficient implantation or stability of protective members in the trout microbiota.

The high genetic variability of *F. columnare* and its broad host range constitute an important limitation for the identification of effective probiotics against this widespread pathogen. Several probiotic candidates isolated from the host provided partial protection against *F. columnare* infection in other conventional fish species such as walleye (*Sander vitreous*) and brook char (*Salvelinus fontinalis*) [51,52]. However, high variability in protection provided by probiotic strains against *F. columnare* was observed depending on the fish batch used, indicating a resistance directly dependent on the fish host genetics [51] or immunological status. Here we reduced this variability using GF and gnotobiotic trout larvae and demonstrated the ability of *Flavobacterium sp.* strain 4466 isolated from Conv trout larvae microbiota to protect against *F. columnare* infection. Furthermore, this bacterium, but not its supernatant, inhibits *F. columnare* growth *in vitro*, which suggests a direct interaction between *Flavobacterium* sp. strain 4466 and *F. columnare*. Intriguingly, *Flavobacterium* sp. strain 4466 encodes a complete subtype T6SS[iii], a molecular mechanism that delivers antimicrobial effector proteins upon contact with target cells and is unique to the phylum *Bacteroidetes* [53]. The members of *Flavobacterium* genus are ubiquitous inhabitants of freshwater and marine fish microbiota and both commensal and pathogenic *Flavobacterium* often share the same ecological niche [54–56]. Whether the *Flavobacterium* sp. strain 4466 T6SS[iii] contact-dependent killing system contributes to colonization resistance by inhibiting *F. columnare* Fc7 growth is currently under investigation. We cannot, however, exclude other mechanisms such as competition for nutrients or

pathogen exclusion upon direct competition for adhesion to host tissues. This process has been suggested for infected zebrafish with efficient colonization of highly adhesive probiotic strains and enhanced life expectancy [24,57,58].

For the past 30 years, the fish farming industry has dedicated considerable efforts to identify probiotic microorganisms for rainbow trout, including Gram-positive and Gram-negative bacteria and yeast [59]. However, the high interindividual and seasonal variability of trout microbiota [60,61] and the random or time-limited colonization ability of exogenous microorganism rarely enables consistent probiotic efficacy. Despite some studies of rainbow trout proposing different endogenous bacterial strains as probiotic candidates, few have demonstrated protective properties against pathogenic bacteria *in vivo* [62–65]. Short-residing probiotics may limit unintended consequences to the microbial community and host system, but the use of endogenous residents may stably modulate the community and protect the fish against reoccurring disease outbreaks over longer timescales [66,67]. The probiotic efficacy of *Flavobacterium* sp. strain 4466 against different strains of *F. columnare* from different fish hosts and geographical origins, suggests that it could be used as a broad probiotic to prevent infections.

In conclusion, we showed that germ-free and gnotobiotic trout larvae enable effective experimental study of microbiota-determined sensitivity to major salmonid freshwater pathogens, leading to the identification of endogenous and exogenous potential probiotic strains. This approach will be instrumental in studying the molecular basis of probiosis against fish pathogens as well as host-pathogen mechanisms, ultimately contributing to the mitigation of rainbow trout diseases in aquaculture.

## Materials and methods

### Ethics statement

All animal experiments described in the present study were conducted at the Institut Pasteur according to European Union guidelines for handling of laboratory animals (http://ec.europa.eu/environment/chemicals/lab_animals/home_en.htm) and were approved by the Institut Pasteur institutional Animal Health and Care Committees under permit # dap200024.

### Handling of rainbow trout larvae

Rainbow trout (AQUALANDE breeding line) "eyed" eggs of 210 to 230 degree-days (21–23 days after fertilization at 10˚C) (dd) were obtained from Aqualande Group trout facility in Pissos, France. Upon arrival, the eggs were progressively acclimatized to 16˚C before manipulation. All procedures were performed under a laminar microbiological cabinet and with single-use disposable plastic ware. Eggs were kept in 145 x 20 mm Petri dishes with 75 mL autoclaved dechlorinated tap water at 16˚C until hatching. After hatching, fish were transferred and kept in 250 mL vented cap culture flasks in 100 mL sterile tap water at 16˚C. Fish were fed starting 21 days post-hatching with gamma-ray sterilized fish food powder every 48 h, 30 minutes before water renewal of half the volume of sterile tap water to avoid waste ($NH_4^+$, $NO_2^-$, $NO_3^-$) accumulation and oxygen limitation.

### Sterilization and raising of germ-free rainbow trout

The eyed rainbow trout eggs received at 210–230 dd were transferred to sterile Petri dishes (140 mm diameter, 150 eggs/dish) and washed twice with a sterile methylene blue solution (0.05 mg/ml) in autoclaved dechlorinated tap water, previously filtered with 0.22 μm system. The eggs, kept in 75 ml of methylene blue solution, were then exposed to a previously described antibiotic cocktail [24] (750 μl penicillin G (10,000 U/ml), streptomycin (10 mg/ml);

300 μl of filtered kanamycin sulfate (100 mg/ml) and 75 μl of the antifungal drug amphotericin B solution (250 μg/ml)) for 24 hours by gentle agitation at 16˚C. Eggs were then washed 3 times with fresh sterile water and treated with bleach (0.005%) for 15 minutes. Following 3 washes with sterile water, eggs were treated for 10 minutes with 10 ppm Romeiod (COFA, France), a iodophor disinfection solution. Finally, eggs were washed 3 times and kept in a class II hood at 16˚C in 75 ml of sterile water supplemented with the previously mentioned antibiotic cocktail until hatching spontaneously 5 to 7 days following the disinfection process. Once hatched, fish were immediately transferred to 75 cm³ vented cap culture flasks containing 100 ml of fresh sterile water without antibiotics (12 larvae/flask). The hatching percentage was determined by comparing the number of hatched and alive larvae relative to the total number of eggs in the Petri dish. Conventionally raised eggs followed the same procedure from their acclimatization until hatching without exposing them to the sterilization protocol mentioned above.

*Sterility*: Sterility was monitored by culture-based and 16S rRNA PCR-based tests at 24 h, 7- and 21-day post-treatment. After feeding started, 50 μl of GF fish flask water was sampled before each water change as well as one larva every week to perform culture-based and 16S rRNA-based PCR sterility tests. 50 μl of rearing water from each flask was plated on LB, YPD and TYES agar plates, all incubated at 16˚C under aerobic conditions. Fish larvae were also checked for bacterial contamination every week using the following methods. Randomly chosen fish were sacrificed by an overdose of filtered tricaine methane sulfonate solution (tricaine, Sigma, 300 mg/L). Whole fish were mechanically disrupted in Lysing Matrix tubes containing 1 ml of sterile water and 425–600 μm glass beads (Sigma). Samples were homogenized at 6.0 m s⁻¹ for 45 s on a FastPrep Cell Disrupter (BIO101/FP120 QBioGene) and serial dilutions of the homogenized solution were plated on LB, YPD and TYES agars. When water samples or fish homogenates showed any bacterial CFU on the different culture media used, the corresponding animals (or flasks) were removed from the experiment. The absence of any contamination in the fish larvae was further confirmed by PCR as follows. Total bacterial DNA was extracted from fish homogenate sample using QIAmp DNA Microbiome Kit (Qiagen) following manufacturer instructions. All reagents used were molecular grade and supplied by Sigma-Aldrich (UK). To detect the presence of microbial DNA, universal specific primers for the chromosomal 16S rRNA (27F: 5'-AGAGTTTGATCCTGGCTCAG-3'; 1492R 5'-GGTTACCTTGT-TACGACTT-3') following the protocol described in [68], the presence of a band of ~1400 bp on an agarose gel indicated contamination and the flask was removed from the experiment.

## Bacterial strains and growth conditions

*F. columnare* strains Fc7 [69], Ms-Fc-4 [70] and IA-S-4 [71] (genomovar I), ALG-00-530 [72] (genomovar II), and *Chryseobacterium massiliae* [34] were grown in tryptone yeast extract salts (TYES) broth [0.4% (w/v) tryptone, 0.04% yeast extract, 0.05% (w/v) MgSO₄ 7H₂O, 0.02% (w/v), CaCl₂ 2H₂O, 0.05% (w/v) D-glucose, pH 7.2] at 150 rpm and 18˚C. *F. psychrophilum* strain THCO2-90 was grown in TYES broth at 150 rpm and 18˚C. *Yersinia ruckeri* strain JIP 27/88 was grown in Luria-Bertani (LB) medium at 150 rpm and 28˚C. *V. anguillarum* strain 1669 was grown in tryptic soy broth (TSB) at 150 rpm and 28˚C. *L. garvieae* JIP 28/99 was grown in brain heart infusion (BHI) broth at 150 rpm and 28˚C. When required, 15 g/L of agar was added to the broth media to obtain the corresponding solid media. Stock cultures were preserved at -80˚C in the respective broth media supplemented with 15% (vol/vol) glycerol.

## Fish infection challenge

Pathogenic bacteria were grown in suitable media at different temperatures until advanced stationary phase. Then, each culture was pelleted (10,000 rpm for 5 min) and washed once in

sterile water. Bacteria were resuspended in sterile water and added to culture flasks at a final concentration of $10^7$ CFU/ml. After 24 hours of incubation with pathogenic bacteria at 16˚C, fish were washed three times by water renewal. Bacterial concentrations were confirmed at the beginning and at the end of the immersion challenge by plating serial dilutions of water samples on specific medium for each pathogen. Ten to twelve larvae were used per condition and experiment and each experiment was repeated at least twice. Virulence was evaluated by daily monitoring of fish mortality up to 10 days post-infection.

## Characterization of culturable conventional rainbow trout microbiota

To identify the species constituting the cultivable microbiota of Conv trout larvae, 3 individuals were sacrificed with an overdose of tricaine at 35 dph, homogenized following the protocol described above and serial dilutions of the homogenates were plated on TYES, LB, R2A and TS agars. The plates were incubated a 16˚C for 48 to 72 hours. All morphologically distinct colonies (based on form, size, color, texture, elevation and margin) were then isolated and conserved at -80˚C in respective broth medium supplemented with 15% (vol/vol) glycerol.

In order to identify individual morphotypes, individual colonies were picked for each morphotype from each agar plates, vortexed in 200 μl DNA-free water and boiled for 20 min at 90˚C. Five μl of this bacterial suspension was used as template for colony PCR to amplify the 16S rRNA gene with the universal primer pair 27f and 1492R. 16S rRNA gene PCR products were verified on 1% agarose gels, purified with the QIAquick PCR purification kit and two PCR products for each morphotype were sent for sequencing (Eurofins, Ebersberg, Germany). Individual 16S rRNA- gene sequences were compared with those available in the EzBioCloud database [73]. A whole genome-based bacterial species identification was performed for *Flavobacterium* sp. strain 4466 with the TrueBac ID system (v1.92, DB:20190603) (https://www. truebacid.com/) [74]. Species-level identification was performed based on the algorithmic cut-off set at 95% ANI or when the 16S rRNA gene sequence similarity was >99%.

## Whole genome sequencing

Chromosomal DNA of *Flavobacterium* sp. strain 4466 isolated from rainbow trout larvae microbiota was extracted using the DNeasy Blood & Tissue kit (QIAGEN) including RNase treatment. DNA quality and quantity was assessed on a NanoDrop ND-1000 spectrophotometer (Thermo Scientific). DNA sequencing libraries were made using the Nextera DNA Library Preparation Kit (Illumina Inc.) and library quality was checked using the High Sensitivity DNA LabChip Kit on the Bioanalyzer 2100 (Agilent Technologies). Sequencing clusters were generated using the MiSeq reagents kit v2 500 cycles (Illumina Inc.) according to manufacturer's instructions. DNA was sequenced at the Mutualized Platform for Microbiology at Institut Pasteur by bidirectional sequencing, producing 2 x 150 bp paired-end (PE) reads. Reads were quality filtered, trimmed and adapters removed with fastq-mcf [75] and genomes assembled using SPAdes 3.9.0 [76].

## Phylogenomic analysis

The proteomes for the 15 closest *Flavobacterium* strains identified by the ANI analysis were retrieved from the NCBI RefSeq database (S2 Table).

These sequences together with the *Flavobacterium* sp. strain 4466 proteome were analyzed with Phylophlan (version 0.43, march 2020) [77]. This method uses the 400 most conserved proteins across the proteins and builds a Maximum likelihood phylogenetic tree using RAxML (version 8.2.8) [78]. Maximum likelihood tree was boostrapped with 1000 replicates.

### Germ free rainbow trout microbial re-conventionalization

Each isolated bacterial species was grown for 24 hours in suitable medium at 150 rpm and 20˚C. Bacteria were then pelleted, washed twice in sterile water and diluted to a final concentration of $5.10^7$ CFU/ml. At 22 dph, GF rainbow trout were mono-re-conventionalized by adding 1 ml of each bacterial suspension per flask ($5.10^5$ CFU/ml, final concentration). In the case of fish re-conventionalization with bacterial consortia, individual bacterial strains were washed, then mixed in the same aqueous suspension, each at a concentration of $5.10^7$ CFU/ml. The mixed bacterial suspension was then added to the flask containing GF rainbow trout as previously described. In all cases, fish re-conventionalization was performed for 48 h and the infection challenge with *F. columnare* was carried out immediately after water renewal.

### Histological analyzes

Histological sections were used to compare microscopical lesions between GF and Conv fish following infection with *F. columnare*. Sacrificed animals were fixed for 1 week in 4% methanol-free paraformaldehyde. Whole fixed animals were then dehydrated in a graded series of ethanol solutions (4 hours in 80% ethanol solution; 4 hours in 95% ethanol solution, and 4 hours in 100% ethanol). Final dehydration was performed by 100% xylene solution $2 \times 4$ hours. Then, samples were embedded in paraffin wash solution (3 x 4 hours) and embedded in paraffin wax ($3 \times 4$ hours) for polymerization.

Semi-thin sections (thickness 5 μm) were cut with a "Leica Ultracut UCT" ultramicrotome (Leica Microsysteme GmbH, Wien, Austria), and mounted on adhesive slides (Klinipath-KP-PRINTER ADHESIVES). Paraffin-embedded sections were deparaffinized and stained with Alcian Blue (AB) and Periodic-Acid Schiff (PAS) to observe both neutral and acidic mucins and Goblet cells quantification. All slides were scanned with the Pannoramic Scan 150 (3D Histech) and analyzed with the CaseCenter 2.9 viewer (3D Histech). Goblet cells quantification was estimated by manual counting of total AB positive cells in blue per villi of the posterior gut.

### Whole fish clearing and 3D imaging

For a 3D imaging of cleared whole fish, fish were fixed with 4% formaldehyde in phosphate-buffered saline (PBS) overnight at 4˚C. Fixed samples were rinsed with PBS. To render tissue transparent, fish were first depigmented by pretreatment in SSC 0.5X twice during 1 hour at room temperature followed by an incubation in saline sodium citrate (SSC) 0.5X + KOH 0.5% + $H_2O_2$ 3% during 2 hours at room temperature. Depigmentation was stopped by incubation in PBS twice for 15 minutes. Fish were then post-fixed with 2% formaldehyde in PBS for 2 hours at room temperature and then rinsed twice with PBS for 30 min. Depigmented fish were cleared with the iDISCO+ protocol [79]. Briefly, samples were progressively dehydrated in ascending methanol series (20, 40, 60 and 80% in $H_2O$, then twice in 100% methanol) during 1 hour for each step. The dehydrated samples were bleached by incubation in methanol + 5% $H_2O_2$ at 4˚C overnight, followed by incubation in methanol 100% twice for 1 hour. They were then successively incubated in 67% dichloromethane + 33% methanol for 3 hours, in dichloromethane 100% for 1 hour and finally in dibenzylether until fish became completely transparent. Whole sample acquisition was performed on a light-sheet ultramicroscope (LaVision Biotec, Bielefeld, Germany) with a 2X objective using a 0.63X zoom factor. Autofluorescence was acquired by illuminating both sides of the sample with a 488 nm laser. Z-stacks were acquired with a 2 μm z-step.

## Agar overlay assay for growth inhibition detection

The growth inhibitory effect of *Flavobacterium* sp. 4466 has been evaluated using an agar spot test. Briefly, 125 μl from an overnight culture of different strains of *F. columnare* adjusted to OD 1 were mixed to 5 ml of top agar (0.7% agar) and overlaid on plates of TYES agar. Five μL of overnight culture of *Flavobacterium* sp. 4466 were then spotted on the overlay of targeted bacteria. The plates were incubated at 28˚C for 24 hours. Growth inhibition of *F. columnare* was recorded by observation of a clear halo surrounding *Flavobacterium* sp. colony. Sterile TYES broth was used as a mock and the experiment were performed in triplicate.

## Statistical methods

Statistical analyses were performed using unpaired, non-parametric Mann-Whitney test or ANOVA using Kruskal-Wallis multiple comparison tests for average survival analysis and the log rank (Mantel-Cox) test for Kaplan–Meier survival curves. Analyses were performed using Prism v8.2 (GraphPad Software). A cut-off of p-value of 5% was used for all tests. * $p < 0.05$; ** $p < 0.01$; *** $p < 0.001$, **** $p < 0.0001$.

## Supporting information

**S1 Table. Flavobacterium sp. strain 4466 taxonomic identification based on genomic similarities.**
(PDF)

**S2 Table. *Flavobacterium* species genomes retrieved from public databases.**
(PDF)

**S1 Fig. Sterility test of rainbow trout larvae raised under GF and Conv conditions.**
(PDF)

**S2 Fig. Growth performance of rainbow trout larvae raised under GF and Conv conditions.**
(PDF)

**S3 Fig. Anatomical comparison of the gut of Conv and GF rainbow trout larvae.**
(PDF)

**S4 Fig. Survival of re-conventionalized trout larvae against *F. columnare* Fc7 infection.**
(PDF)

**S5 Fig. Histological comparison of the gut of infected and non-infected Conv and GF rainbow trout larvae.**
(PDF)

**S6 Fig. Phylogenetic tree illustrating the relationship between *Flavobacterium* sp. strain 4466 and the closest 15 *Flavobacterium* species based on ANI analysis.**
(PDF)

## Acknowledgments

We thank Rebecca Stevick, Jean-Pierre Levraud, Pierre Boudinot, Eric Duchaud, Mark McBride and Christophe Beloin for critical reading of the manuscript. We thank Laurent Debruyne and Jérémie Vieville from the Pissos Aqualande Trout breeding station. We are grateful to Jean-François Bernardet and Mark McBride for kindly providing us some of pathogenic microorganisms used in this study. We thank Rustem Uzbekov for his help in histology analyses performed in the context of a service provided by the IBiSA Microscopy facility, Tours

University, France. iDISCO imaging was established and performed by Christelle Langevin and Maxence Fretaud (INRAE EMERG'IN IERP phenotyping platform) and light-sheet images were acquired at the Institut de la Vision. This work has benefited from the facilities and expertise of @BRIDGe (Université Paris-Saclay, INRAE, AgroParisTech, GABI, 78350 Jouy-en-Josas, France).

## Author Contributions

**Conceptualization:** David Pérez-Pascual, Jean-Marc Ghigo.

**Data curation:** David Pérez-Pascual, Rafael Patiño-Navarrete, Jean-Marc Ghigo.

**Formal analysis:** David Pérez-Pascual, Jean-Marc Ghigo.

**Funding acquisition:** Jean-Marc Ghigo.

**Investigation:** David Pérez-Pascual, Sol Vendrell-Fernández, Bianca Audrain, Joaquín Bernal-Bayard, Jean-Marc Ghigo.

**Methodology:** David Pérez-Pascual, Sol Vendrell-Fernández, Bianca Audrain, Joaquín Bernal-Bayard, Rafael Patiño-Navarrete, Jean-Marc Ghigo.

**Project administration:** David Pérez-Pascual, Jean-Marc Ghigo.

**Resources:** David Pérez-Pascual, Vincent Petit, Dimitri Rigaudeau, Jean-Marc Ghigo.

**Supervision:** Jean-Marc Ghigo.

**Validation:** David Pérez-Pascual, Jean-Marc Ghigo.

**Visualization:** David Pérez-Pascual, Sol Vendrell-Fernández, Jean-Marc Ghigo.

**Writing – original draft:** David Pérez-Pascual, Jean-Marc Ghigo.

**Writing – review & editing:** David Pérez-Pascual, Sol Vendrell-Fernández, Bianca Audrain, Joaquín Bernal-Bayard, Vincent Petit, Dimitri Rigaudeau, Jean-Marc Ghigo.

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
