## [Decision Letter · Decision Letter 0]

14 Aug 2020

Dear Prof. GHIGO,

Thank you very much for submitting your manuscript "Gnotobiotic rainbow trout (Oncorhynchus mykiss) model reveals endogenous bacteria that protect against Flavobacterium columnare infection" for consideration at PLOS Pathogens. As with all papers reviewed by the journal, your manuscript was reviewed by members of the editorial board and by several independent reviewers. In light of the reviews (below this email), we would like to invite the resubmission of a significantly-revised version that takes into account the reviewers' comments. 

We cannot make any decision about publication until we have seen the revised manuscript and your response to the reviewers' comments. Your revised manuscript is also likely to be sent to reviewers for further evaluation.

Sincerely,

Karen Guillemin

Guest Editor

PLOS Pathogens

Nina Salama

Section Editor

PLOS Pathogens

Kasturi Haldar

Editor-in-Chief

PLOS Pathogens

orcid.org/0000-0001-5065-158X

Michael Malim

Editor-in-Chief

PLOS Pathogens

orcid.org/0000-0002-7699-2064

Reviewer's Responses to Questions

**Part I - Summary**

Reviewer #1: Strengths: directly studies trout, which are an economically important but under utilised experimental system. Logical experiments establish a system using gnotobiotic larvae to model antibiotic-associated opportunistic infections, demonstrate protection by cocktail and then individual species of normal microbiome.

Well written and presented. Missing a few connections (ie CFU recovery) that would strengthen the manuscript by providing additional readouts. Missing mechanistic insight into the mode of protection at this stage but I believe the authors could use the system to rapidly establish a mechanism of protection.

Reviewer #2: This manuscript describes the generation of a gnotobiotic protocol for rainbow trout, and the ability of the trout microbiota to protect against pathogenic bacteria. The paper also describes use of a mock community of trout microbiota that conferred resistance to the infection, and that monoassociations and mock community exposures indicate a specific member of the microbiota produces anti-bacterial activity against the pathogen. This is a significant finding both for trout aquaculture and for gnotobiotic studies in general. As far as I can tell, similar studies have not been done in trout. In general, these experiments were executed well, although I do have some concerns, most of which may be clarified with some editing.

**Part II – Major Issues: Key Experiments Required for Acceptance**

Reviewer #1: Figure 2 and SFig 4

Need to have at least some basic microscopic analysis of histology. Eg Alcian blue staining for intestinal goblet cell enumeration. Additionally, the SFig4 uninfected samples do not appear well matched.

SFig 4B: there appear to be significant signs of IEC vacuolation and, potentially, mucus staining abnormalities in the CONV-infected (there's something in the lumen). This concern needs to be addressed by higher resolution imaging of more appropriately-stained tissue (Alcian blue, H&E etc)

Figure 4 onwards "mean death day" plots need to be complemented by CFU recovery or bacterial 16s DNA/RNA quantification.

3. Mechanistic evidence

The in vitro data in SFig5 is central to the mechanism of the manuscript and needs to be included as a full figure. The in vitro data should be complemented by in vivo CFU recovery showing Flavobacterium colonisation prevents/reduces colonisation of the GI tract by F. columnare.

Lines 299-303 suggest mechanism that could be investigated. This paper would be significantly strengthened by the demonstration of at least one of these mechanisms in vivo.

Reviewer #2: Supplement figure 5: Was this done in triplicate and these are representative? If so, indicate. If not, these should be repeated.

**Part III – Minor Issues: Editorial and Data Presentation Modifications**

Reviewer #1: Line 120: Is hatching efficiency directly related to embryo survival or could the increased hatching of GF embryos be due to weakened chorions?

Line 145: 10^7 CFU/ml is a lot of bacteria. Was the mortality rate dose responsive? Surprising that V. anguillarium did not kill larvae.

Figure 4 "mean death day" plots: Please show survival curves too for completeness. Ok as supplementary Fig.

Line 174: No crucial, but do the Mix11 species colonise at equal proportions? Is there a hint (that could be tied into later observations) that Flavobacteria outperform the other species?

Sentence in lines 281-283 does not seem related to the rest of the paragraph.

Line 292: highlight protection against "multiple" strains of F. columnare in zebrafish embryos.

Related note: the switch to zebrafish embryos for demonstrating protection against multiple pathogen strains was not well justified. Paper would be strengthened by showing protection of trout larvae against multiple pathogenic strains. Is this omission patent related?

T tests are not appropriate for the multiple comparisons performed in Fig 4 and 5. Reanalyse with ANOVA

Reviewer #2: Minor points:

Introduction:

90: “sterile” indicates the fish in those studies had no living organisms. Reference 27 uses antibiotics to kill microbes and only examined growth in aerobic, high nutrient conditions, not by 16S PCR. Check other references for sterility confirmation, or change “sterile conditions” to “reduced microbial load” or “reduced bacterial load”

Methods:

Line 335: Were these all from the same clutch of fish, or several different clutches? Given the reference to host genetic background on the ability of the microbiota to protect against pathogens, this could be significant.

Line 340: kept at 16C while in petri dishes? Add that detail, or in line 338 that fish were maintained at 16C until the end of the experiment (or whenever their temp was changed).

Also line 340: Is the water tap water, ocean water, what salinity? More details needed.

Lines 341, 348 and elsewhere: how was the water and methylene blue solution sterilized? What type of water was used (instant ocean, tap water, RO water? Is it the autoclaved dechlorinated water from line 340?)

Line 343: specify sterile water was added.

Line 347: this is at odds with line 335, which says fish were collected at 210-230 degree days. Were only the 210 dd fish derived germ free?

379: add “Following the protocol in [68]. Presence of a band of X bp on an agarose gel indicated contamination and the flask was removed from the experiment.”

401: Is that 10-12 fish per flask, or is that 5-6 fish in each flask and each flask is in triplicate? This relates to the issues with replication.

Need to include a method for the microbe-microbe interaction studies.

Results:

In some places the manuscript states fish were reconventionalized at 21 dph, others it’s 22 dph. Be consistent.

Figure 1: Label the timeline with “dph”

Table 1: add the identification of the strains so that if people contact you to get the strains, they can give you the correct strain identification. For example, if I contacted you and asked for Aeromonas rivipollensis 1, how would you ensure that I received the strain from this paper? Related, indicate whether you deposited the strains to a repository.

Supplement figure 1 and elsewhere: Is figure S1 a representation of the PCR gels used, or is that all the flasks that were used in these experiments? It appears that there were 16 germ free flasks; if the pathogen exposures were done in triplicate flasks of 10 fish each (not sure how you would get the error bars in figures 3 & 4 without that) that would be 15 of the 16 flasks, not enough flasks left for the Re-conv, mix 11, etc experiments. State that the PCR gel is representative (if it is) of the PCR results and clarify how many flasks were used for each experiment.

Supplement figure 4: Can’t read the scale bars to determine whether the images are at the same scale.

Supplement figure 5: Scale bars?

Line 188: “Consistently” – how many times was the sequencing performed and/or reanalyzed? This just seems like an odd word here.

Discussion:

Line 242: Your fish were not GF up to 35 dph. They were GF from -5dph to 35 dph. As far as I can tell, your fish were conventional for 210-230 degree days before they were treated. Fish undergo major development while in their corions.

Major concerns:

It is not clear what the biological replication of the experiments are. For example, in the pathogen exposures: how many flasks of fish were exposed to the pathogens? For examples of this, see line 916, 144-148, 159-161, 171-175. Line 402 indicates that each experiment was repeated at least twice; does that mean from derivation of germ free fish to termination of the experiment, or does that mean the germ free fish were divided and at least two flasks were used in each experiment?

Lines 120-122 and elsewhere: Were conventional eggs made germ free and then microbes were added, or were they untreated (that is, didn’t go through the germ free derivation protocol)? If the latter, the process of deriving eggs germ free may have affected the fish in lines 120-122 and elsewhere.

Line 121-122: “Possibly due to higher susceptibility of Conv eggs to….” This is speculative. A number of things can effect hatching efficiency.

Regarding the iDISCO, 3D imaging, and histology (described in results in 131-138 and histological results in lines 153-154): What measurements were taken? What were the results from the measurements? Was the person doing the analysis blind to the treatment? “No damage” doesn’t speak to what aspects of inflammation/lesions/delayed growth/color, etc, was measured and used to determine there was no damage or delay in development.

Lines 339-340: Were the petri dishes with treated fish sealed to ensure that microbes weren’t able to get into the petri dishes? Fungi in particular are able to get into unsealed petri dishes. These would not be detected by plating at 16C for 2 days or by 16S sequencing.

The methods for the germ free derivation and the fish handling are confusing. I would recommend combining the two into one section called “Sterilization and husbandry of fish”.

Line 916 and 144-148: Was the water tested after the exposure and at the end of the experiment to ensure these were monocultures? Line 401 indicates specific media was used for each pathogen; if these are selective media, other microbes that contaminate the water/fish would not be detected. Yersinia ruckeri, V. anguillarum, and L. garvieae were all grown at 28C. F. columnae, Chryseobacterium massiliae and F. psychromphilum were grown at 18C. It could be assumed that most of the general microbiota and other microbes that may contaminate the flasks grow at 16C. These temperature differences may inhibit the ability to detect contaminants in the cultures with the pathogens or other microbe exposures.

PLOS authors have the option to publish the peer review history of their article (what does this mean?). If published, this will include your full peer review and any attached files.

Reviewer #1: **Yes: **Stefan Oehlers

Reviewer #2: No
---

## [Decision Letter · Decision Letter 1]

24 Dec 2020

Dear Prof. GHIGO,

We are pleased to inform you that your manuscript 'Gnotobiotic rainbow trout (Oncorhynchus mykiss) model reveals endogenous bacteria that protect against Flavobacterium columnare infection' has been provisionally accepted for publication in PLOS Pathogens, pending some textual changes requested by the reviewers. 

Before your manuscript can be formally accepted you will need to make some textual changes to address issues raised by the reviewers, as detailed in the comments below. 

Best regards,

Karen Guillemin

Guest Editor

PLOS Pathogens

Nina Salama

Section Editor

PLOS Pathogens

Kasturi Haldar

Editor-in-Chief

PLOS Pathogens

orcid.org/0000-0001-5065-158X

Michael Malim

Editor-in-Chief

PLOS Pathogens

orcid.org/0000-0002-7699-2064

Reviewer Comments (if any, and for reference):

Reviewer's Responses to Questions

**Part I - Summary**

Reviewer #1: The authors have addressed all of my comments with new data or in their direct response.

Reviewer #2: This is a resubmission of a manuscript I previously reviewed. This revised version is significantly improved as the authors have address most of the previous reviewer's concerns. As I said previously, this paper describes the generation of a gnotobiotic protocol for rainbow trout, and the ability of the trout microbiota to protect against pathogenic bacteria. The paper also describes use of a mock community of trout microbiota that conferred resistance to the infection, and that monoassociations and mock community exposures indicate a specific member of the microbiota produces anti-bacterial activity against the pathogen. This is a significant finding both for trout aquaculture and for gnotobiotic studies in general. As far as I can tell, similar studies have not been done in trout. In general, these experiments were executed well, although I do have some concerns that may be clarified with some editing.

**Part II – Major Issues: Key Experiments Required for Acceptance**

Reviewer #1: No further experiments are requested.

Reviewer #2: “Sterile” is incorrectly used in this manuscript. “Sterile” and “sterilized” indicates the fish in those studies had no living organisms. The fish and water were not tested for viral material or for fungi. Fungi will not necessarily grow under the conditions described for sterility testing (16C for several days on three types of media) and absence of fungi was not confirmed by PCR/sequencing. Please use “bacteria-free”, as is standard in fish gnotobiotic studies in which the above have not been performed. Germ free is also fine, as this generally indicates bacteria free.

**Part III – Minor Issues: Editorial and Data Presentation Modifications**

Reviewer #1: Extensive proof reading is required.

Reviewer #2: Line 90: I checked several of the references. Most do not claim to derive fish as sterile (no microorganisms) but instead refer to their systems as “bacteria-free”. In the references I checked, and in most papers I have read on this topic, sterile is used to describe the water the fish are in (as in the autoclaved or filtered water) or the tools used, but the fish themselves, especially if they are treated with antibiotics but not soaked in bleach or PVPI, are considered “bacteria-free” or “reduced microbial load”. This is especially true for studies which don’t test specifically for virus or fungal species, which includes most fish studies. Change the word “sterile” to “bacteria-free” or something similar. Also, the cod reference (Forberg et al 2011) was fine and did not need to be removed.

Line 116-119: Hatching timing does not indicate viability. These are two separate measurements. Revise line 119 to indicate that you measuring timing to hatch.

Line 120-123 belong in discussion, not results.

Figure 3: axis should be labeled Time (days post exposure)

Figure S5: Panel A, left side images of GF and CV are in different orientations. Panel B, GF and Conv are in different parts of the plane of the fish, making comparisons difficult. In the right figure of panel B, the bottom image is either a different part of the gut or something else is going on because the folds are missing from the top of the image (on a closer examination, that may be the lumen of the gut. Indicate in the left images where the zoomed in images are from, if they are from the images on the left of the same panel). The folds in the infected CV fish also look smaller and more spaced out than the folds in the GF fish in the right side images of panel B. This may be due to the location in the gut that the image was taken, but that also appears to the be the case in the left images.

Figure S5: Scale bars are needed.

PLOS authors have the option to publish the peer review history of their article (what does this mean?). If published, this will include your full peer review and any attached files.

Reviewer #1: **Yes: **Stefan Oehlers

Reviewer #2: No

---

## [Editor Report · Acceptance letter]

24 Jan 2021

Dear Prof. GHIGO,

We are delighted to inform you that your manuscript, "Gnotobiotic rainbow trout (Oncorhynchus mykiss) model reveals endogenous bacteria that protect against Flavobacterium columnare infection," has been formally accepted for publication in PLOS Pathogens.

Best regards,

Kasturi Haldar

Editor-in-Chief

PLOS Pathogens

orcid.org/0000-0001-5065-158X

Michael Malim

Editor-in-Chief

PLOS Pathogens

orcid.org/0000-0002-7699-2064